# The role of project management office in the implementation of strategic plans in project-based organisations

**Maqsood Ahmad Sandhu**[1], **Tareq Al Ameri**[2], **Asjad Shahzad**[3]*, **Afshan Naseem**[3]

**1** College of Business and Economics, United Arab Emirates University, Al Ain, UAE, **2** Infrastructure and Facilities Office, ADEK- HQ, Abu Dhabi, UAE, **3** Department of Engineering Management, National University of Sciences and Technology, Islamabad, Pakistan

☯ These authors contributed equally to this work.
\* asjad.shahzad@ceme.nust.edu.pk

**Data Availability Statement:** All relevant data are within the paper and its Supporting Information files.

## Abstract

The role of the project management office (PMO) in improving project execution has recently been acknowledged and is gaining popularity in project-based organizations to furnish various options for project-solving approaches. This study aims to identify and test the ability of PMO roles in implementing the strategic plan of the organization. This research adopted survey-based quantitative research. The questionnaire was shared with 450 staff members working in 19 project-based organizations. 268 usable questionnaires were received. The methodologies for the development of project management, monitoring and controlling project performance, organizational learning, monitoring and controlling project performance, and improving organization structure and communication were the top five PMO roles involved in the execution of strategic plans, according to the results. At the same time, the criteria of twelve top metrics were recognized to determine the effectiveness of the PMO department. This study has research implications for the researchers involved in the exploration of the specific benefits of PMO.

## 1. Introduction

The emergence of cutting-edge technologies and modern management strategies in contemporary business and industrial sectors has notably heightened the complexity involved in overseeing the diverse stages of project implementation. This complexity has ingrained itself within management paradigms, representing a critical factor in project-related issues [1]. The complex nature of project environments has led to unfavourable challenges for numerous project-centric organizations such as resource scarcity, inconsistencies in management processes and methodologies, inadequate coordination among concurrent projects, and improper project selection [2].

The project business environment has created new research issues that need to be discussed carefully. Consequently, this emerging project landscape has presented businesses with unprecedented management challenges, igniting a strong interest in discovering and

**Funding:** The author(s) received no specific funding for this work.

**Competing interests:** The authors have declared that no competing interests exist.

employing efficient methods and resources to improve the way their strategic plans are carried out through the accomplishment of projects. These efficient methods include leveraging Project Management Office (PMO) functions as impactful contributions in the project environment, which are increasingly seen as crucial success factors for project execution in the present era [3].

In the current era, the majority of private as well as public sector project-based organisations are trying to explore new management tools and strategies that can aid them in streamlining their project execution and implementation [4]. PMO is one such effective tool, which has stemmed from multiple inter-related disciplines, such as business theories, project management, information technology, organisational behaviour, etc. This tool is important in instilling project management practices in an organisation [5]. Therefore, the PMO is considered a regularly developing feature of the project landscape and a dynamic management phenomenon. The survey of Hobbs and Aubry, et al. [6] has concluded: "*the structures, roles, functions, and validity of the PMO vary significantly per the business nature of the hosted organisation, and in the context of project purposes and justification*".

In order to enhance project performance and successful completion, numerous project-based organizations are increasingly embracing innovative management solutions. However, the primary drive behind implementing a Project Management Office (PMO) within these organizations often stems from a dual purpose: to enhance the strategic management of projects and concurrently decrease the occurrence of project failures that fail to meet customer and stakeholder expectations, typically due to budgetary excesses or unacceptable delays [7].

The contribution of each PMO function follows an evolutionary pattern that is determined by how the role changes over time inside the project-based organization. The PMO department adjusts by adding additional functional responsibilities and contributions as it gains experience doing a variety of tasks during the course of a project [8]. Every PMO role is evolving without decreasing the significance of their present roles. However, a directional association has been shown between the increasing efficacy and the consistent rise in the significance of PMO functions provided to the host organization; thus, the latter may experience an increase in its strategic influence [9,10]. It is believed that the PMO is essential to accomplishing the project's goals [11].

Numerous scholarly works have discussed the diverse forms and methodologies of PMO, ranging from historical accounts to contemporary analyses, indicating its evolutionary journey and adaptive functions over time [12]. This demonstrates an expanding corpus of project management research, including a range of theories and applications, that offers convincing proof of the PMO's activities. This might give the PMO the necessary adaptability to confront and overcome a project execution failure, which is more common in today's project-based business environment.

Even while PMOs have been ingrained in the business and industrial worlds for a long time, their activities have continued to vary to the point that there is now no consensus on what defines or characterizes PMO roles and procedures, meaning they remain up for debate [13]. The PMO units are structured according to the requirements of the project-based organisations meaning that PMO is an organisation-specific body. The diverse array of roles within the PMO complicates the process of measuring its value-added contributions, as each role contributes specific value to its host organization in distinct ways [13]. Unless a proper method and acceptable measures determining the PMO values are employed, otherwise, invalid conclusions can be reached about the actual value that a specific PMO contributes. The previous research has predominantly targeted the role of PMO in the operational and tactical aspects of the organization, however, there is a void to specifically address its influence on the project-based organizations' strategic elements [14–16]. Therefore, in order to close this gap, it will be

necessary to identify the precise responsibilities that PMOs play in helping firms implement their strategic plans. In other words, we are interested in finding the answers to the questions

*What are PMO enablers that have a significant influence on the implementation of the organization's strategic plan?*

This research would contribute to the literature by identifying the specific domains that may be improved by adopting the PMO in an organization. Furthermore, this research shall fill the research gap of the unavailability of empirical studies on the role of PMO in executing the strategy in an organization. This will help the project-based organizations to align the planning of the strategy of the firm with the project objectives and outcomes. This shall help the practitioners effectively explore and develop their strategic plan, which in return could assist in achieving organizational goals. Moreover, the study would have significant implications for the hosted organizations, such as utilizing PMO, not only as a means of managing and controlling projects but also as a way to enhance the achievement of the strategic plan. These results can be useful in creating the conceptual model (PMO) that may be tailored to apply to the same PM methodology in different settings. Moreover, it will pave the way for further investigations.

## 2. Literature review and theoretical framework

### 2.1 Importance of PMO for an organization

Project Management Office (PMO) has a significant influence on the strategic performance of an organisation [17]. Multiple studies have been carried out in the past for the investigation of the definitions, responsibilities, and practices inside the framework of strategic planning and the business model of an organisation. Many studies focused on finding a suitable definition of PMO that could reflect the essence of its functions and roles. Project Management Institute (PMI) defines PMO as "*A project management office is a management structure that standardizes the project-related governance processes and facilitates the sharing of resources, methodologies, tools, and techniques*" [18]. Its primary role is to manage and oversee several projects under the umbrella of a single organization [19]. PMO is seen as a valuable organization whose purpose is to provide benefits to the organization [20].

It is still impossible to come up with a single, broadly accepted definition for the PMO, though. Hobbs and Aubry [21] for example identified three elements that could spark a discussion on different definitions: Three things about PMOs: 1) they are a relatively new phenomena; 2) they can take on a wide range of shapes and functions; and 3) there hasn't been a thorough examination of PMOs as organizational units.

According to earlier research, project management offices (PMOs) are a relatively new addition to project-based organizational structures. They have experienced many functional changes in a short amount of time, depending on the PMO unit's start-up point, success, and sustainability [22,23]. "Many PMO roles have an initial short lifespan before they are restructured and their functions refocused," according to a report by Hobbs, et al. [22]. According to these writers, it would be incorrect to draw the conclusion that PMO units only provide a project-based organization with a small amount of long-term benefit.

However, creating a successful PMO unit does not guarantee that sustained PM competencies or expected PM value will be reliably attained [24]. It is crucial to include particular components in the PM process and to routinely participate in particular PMO-related activities so that sustainability of PM competencies and values can be ensured. Project managers and experts are regarded as the primary stakeholders in this scenario, as they are in charge of creating and keeping an effective PMO unit.

Furthermore, an organization's ongoing efforts and concerns ought to be focused on maintaining the most recent stage of project implementation. As a result, it becomes critical to determine any new tactics and processes required for carrying out the strategic strategy effectively. Organizations also need to monitor the efficient execution of PMO roles and develop project management competency to continuously deliver project values.

## 2.2 The strategic role of PMO

The transdisciplinary character of academic research in a PMO presents a significant problem. Researchers in project management domains, for example, have given diverse and particular roles and practices a considerable deal of attention and worry [4,25–27]. Nonetheless, the aforementioned scholars' works highlighted a range of key goals, responsibilities, and procedures that are often used in a large number of project-based organizations.

The various functions of the PMO have equipped it with dynamic flexibility across a broad spectrum of organizational tasks. Hobbs and Aubry [21] identify about twenty-seven functions and roles that PMOs can sufficiently perform. Out of these, twenty-one roles and functions are significant for a minimum of 40% of PMOs in which surveys were conducted, worldwide, and out of these, 7 roles strongly link to the strategic planning of the organisation i.e "Reporting to upper management about the project status", "Coordination between simultaneous and multiple projects", "Promoting a culture of project management within the organization", "Participating and involving in organization's strategic planning "Managing multiple portfolios", "Participating in the selection process of new projects with priority", Managing single or multiple projects"

Because not every PMO is able to carry out every defined role, a PMO's performance varies depending on the organization. Within project-based organizations, learning lessons and professional experience produced from past executed projects, whether succeeded or failed, are acknowledged as helpful resources that can be used in the further improvement of the forthcoming project execution. There has been a significant increase in the attention paid to the delivery of values that are possibly added by various PMO roles in the hosted organisations. PMO is essential to the development and upkeep of an organization's values. A lot of sectors that rely on projects have begun working together to co-create value with their stakeholders and customers. A PMO may provide value to an organization and portfolio of projects, according to Karayaz and Gungor (24). The authors suggested tying the value produced by each of a PMO's distinct tasks to the value provided by the organization in order to quantify the value added by a PMO.

## 2.3 Theoretical research framework

Previous studies on PMO positions highlight the inherent difficulties and complexities of working with various alliances [28]. The current study shows a conceptual framework that demonstrates how the associated variables of a project management office (PMO) collaborate and are interdependent while implementing a strategic plan of a project-based organization.

The theoretical framework that we have proposed is based on the research by Dai and Wells [29] Hobbs and Aubry [21] José, et al. [30] and Philbin [31]. The studies define the specific roles of PMOs in the context of the organization. The framework in this research aims at "*investigating in-depth the involvement of selected PMO roles in the implementation of the project-based organisation's strategic plan*".

The predicted variable in the theoretical framework is the "implementation of the strategic plan of the project-based organization." From the list of frequently used PMO functions provided by Hobbs and Aubry five independent variables were taken out: i) "strategic project

**Table 1. A comparison of PMO roles.**

| Roles | Dai and Wells [29] | Hobbs and Aubry [21] | José, et al. [30] | Philbin [31] |
|---|---|---|---|---|
| Strategic Management | ----- | Participant in PM strategy | ----- | ----- |
| Control and Monitoring | Offering project management support | Controlling/monitoring project performance | Monitoring and controlling project performance | Report project status to upper management and provide advice |
| Methodology/ Competency | Developing/maintaining PM standards; offering consultancy; delivery of training | Developing and promoting PM competencies & methodologies | Develop or select a methodology for project management processes and methods; Implement and operate an enterprise project information system | Develop or select a methodology for project management processes and methods; Implement and operate an enterprise project information system |
| Multi projects | Offering project HR and staffing | Ability to manage and control multi-projects | | Manage one or more projects or programs |
| Organisational Learning | Supporting project documentation and archives | Developing organisational learning and culture | Conduct post-project reviews | Implement and manage a risk database |

management (PM)"; ii) "project controlling and monitoring performance (PMC)"; iii) "creating methodology and competency (MC)"; iv) "multi-projects management (MPM)"; and v) "supporting the process of organizational learning (OL)". Four of these variables (except for the strategic management) are also common in the other three studies, as shown in Table 1.

Two new roles that were not previously explored in the PMO literature (previously discussed) are proposed as independent variables based on a large body of literature and the principles of parsimony and comprehensiveness [32]. These are i) "organizational structure and communication (OSC)" and ii) "sustainability of project values (PV)". In Fig 1, the conceptual framework is displayed.

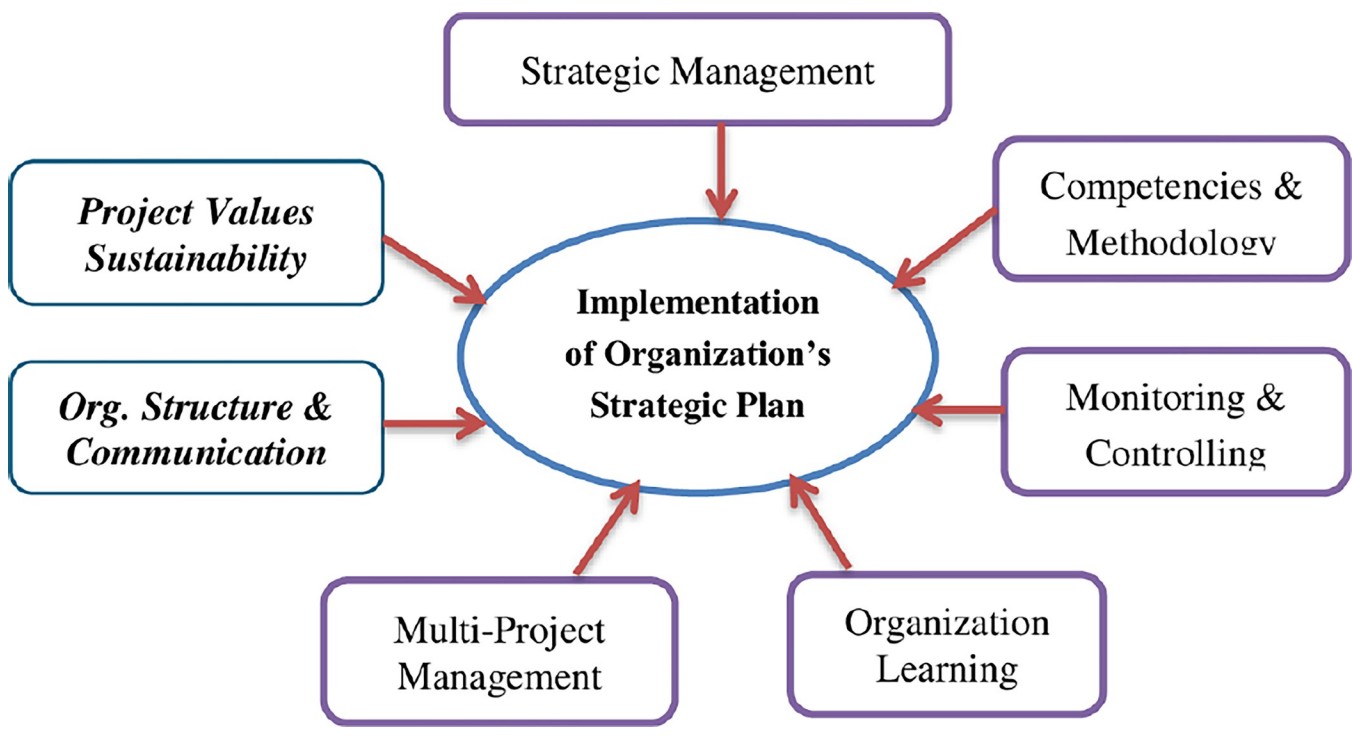

**Fig 1. Conceptual framework.**

## 2.4 Hypotheses

The hypotheses stem from the research questions and theoretical framework. We proposed seven hypotheses. Each of the hypotheses is divided into items or questions that may be negative/ positive to cater to individual roles

### 1) Strategic management (SM) role:

Many times, when project execution happens outside of an organization's traditional administrative boundaries, it calls for specialized strategic management, leadership abilities, coordination techniques, and incentive programs [33]. Although the phrases "strategic management" and "strategic planning" are sometimes used synonymously in literature, strategic management is a broader concept than "strategic planning" since it encompasses not only the creation of strategic plans but also its execution and assessment [34].

Not much research has been addressed and acquired on the implementation of strategic management, despite the fact that these techniques are widely used across the globe [35]. Also, the emotional intelligence aspects of strategic human resource management have been discussed, without full discussion on the implementation of strategy in projects [36]. This variable is adopted by the conceptual framework in order to address its efficacy in carrying out project strategic plans. As a result, this independent variable may effectively improve the organization's capacity to recognize the elements necessary for excellent project management.

Thus, the related hypotheses state:

$H1_o$: *The strategic management role of the Project management office is not associated with the implementation of the strategic plan of a project-based organization.*

$H1_a$: *The strategic management role of the Project management office is associated with the implementation of the strategic plan of a project-based organization.*

### 2) Project management methodology and competency (PMMC) role:

Appropriate competency frameworks have been utilized by a large number of private as well as public companies to define the necessary competencies for each of the important project-related jobs within the company [37]. Within a strategic framework, projects would alter the hosted organization's working conditions in terms of how it operates because they would enable the organization to mobilize its people and resources to produce additional sources of value and give it an advantage in the market [37]. Programs cannot be successfully implemented and run efficiently with just the Project Control-Cycle method. In order to guarantee the sustainability of program management, frameworks such as PRINCE2TM and the PMBOK® provide managers of projects with templates, instructions, and best practice guides. But as every project is different, each project manager's knowledge and skills allow them to customize approaches to fit their unique projects [38].

Following standard project management techniques, offering processes, tools, and other resources, fostering a project culture inside organizations, mentoring, and raising professional knowledge and competency are all aspects of this PMMC function. Associating the outcomes of the project with the strategic goals of organizations can be hypothesized. As a result, associated theories claim:

$H2_o$: *The developing project management competency and methodology role of the project management office is not associated with the implementation of the strategic plan of a project-based organization*

**H2$_a$**: *The developing project management competency and methodology role of the project management office is associated with the implementation of the strategic plan of a project-based organization.*

### 3) Monitoring and controlling performance (MCP) role:

Controlling the project's activities depends on successful interaction at the workplace or project site. The project manager regularly monitors the project control cycle using intra-communication channels, whether from a remote desk or on-site. Better control via communication requires i) consulting project team members, ii) demonstrating the gathered data and information to all project staff, and iii) communicating regularly with the stakeholders, tracking and oversight procedures are therefore considered updating methods.

Pierce Jr [39] noted a number of factors that could possibly lead the proposed assignments' schedules to get behind schedule. These factors include i) contractual date changes, such as extending the time taken to deliver; ii) work sequence changes made by on-site people without intimating their supervisor/manager; and iii) changes in the delivery dates of materials, as a lag may a significantly impact the execution of the project

The MCP's role includes reporting the various functions that relate to each other, such as the status of the project etc. This streamlines the exchange of data, thereby, making the execution of running projects as per the schedule. Thus, the related hypotheses state:

**H3$_o$**: *The monitoring and controlling performance role of the project management office is not associated with the implementation of the strategic plan of a project-based organization*

**H3$_a$:** *The monitoring and controlling performance role of the project management office is associated with the implementation of the strategic plan of a project-based organization*

### 4) Organisational learning (OL) role:

While projects offer chances for further professional development, the practical implementation of learning depends on the organization's overall style of learning. Turner and Keegan [40] conducted an examination of learning through project practices in many European organizations and found that there are three main obstacles to learning in the firms that are based on projects i.e. time (ii) centralization; and (iii) delay. This emphasizes how important it is to learn across organizational boundaries—both inside and across distinct companies. They proposed that boundary objects serve as a sort of "translation" so that several project managers, each with varying opinions and viewpoints about the value and potential applications of the information produced, can utilize the same expertise and information.

Anbari, et al. [41] looked at the connection between the management of knowledge and learning in projects within the framework of reviewing the procedures in projects. They looked at "the reasons why post-project reviews are not conducted frequently in practice, despite being generally thought to be beneficial in the literature." They concluded that the key to successful project execution and, consequently, company competitiveness is the regular gathering of project lessons learned.

This OL role is assumed as a key enabler for developing organisational loyalty and specific experience. It also relates to the post-project reviewing process, auditing tasks, evaluating PMO performance, and

managing the lessons learned and professional experience, risks, documentation, and archive databases. Its related hypotheses state:

**H4$_o$**: *The organisational learning role of the project management office is not related to the implementation of the strategic plan of a project-based organization*

**H4$_a$**: *The organisational learning role of the project management office is related to the implementation of the strategic plan of a project-based organization*

**5) Multi-project management (MPM) role**:

Management of multiple projects simultaneously is not a new concept. However, from the mid-20th century onwards, both professionals and scholars have increasingly focused on studying project management. Therefore, project management has evolved as a separate field over the past three decades [42]. Initially, the projects were handled as separate entities. It was rarely noticed that more than a couple of projects were managed by the organisation over many years.

Various concepts have been proposed to enhance the operation efficacy in the multi-project environment such as establishing a Project management office that is organization-specific [43]. Both private and public sectors have now prioritized the management of multi-projects. This MPM position involves managing the distribution of resources to sustain the progress of concurrent projects by effectively coordinating resource allocation, resolving conflicts, and reducing the chances of overlapping projects. Consequently, its related hypotheses state:

**H5$_o$**: *The multi-project management role of the project management office is not associated with the implementation of the strategic plan of a project-based organization*

**H5$_a$**: *The multi-project management role of the project management office is associated with the implementation of the strategic plan of a project-based organization*

**6) Organisational structure and communication (OSC) role**:

It is essential to comprehend the communication process in the context of an organization, a project, and many projects. Three components make up communication: a receiver, a transmission channel or medium, and a transmitter or sender. Additionally, the codes used to transport a message constitute the media for communication [44]. However, several academic studies recognized the project border "interface" as an important obstacle that prevents the project from communicating with its parent company. As a result, poor communication may cause misconceptions about the objectives and the scope of the project plan. This can lead to inadequately defined tasks and critical processes, and uncertainty regarding the responsibilities of the team members. Nevertheless, ineffective communication may cause the project to fail [44]

The OSC role focuses on implementing efficient tools for PMO operations and communication. Tailoring communication patterns within the organization to meet its needs, the role aims to strengthen channels to project stakeholders, ensure timely updates through information channels, and facilitate project continuity by transferring necessary technology and innovative methods. Hence, its related hypotheses state:

**H6$_o$**: *The organisational structure and communication role of the project management office is not associated with the implementation of the strategic plan of a project-based organization*

**H6$_a$**: *The organisational structure and communication role of the project management office is associated with the implementation of the strategic plan of a project-based organization*

**7) Project value sustainability (PVS) role**:

Establishing sustainable procedures that support innovative thinking and concept feasibility is the first stage in creating project value. This means overseeing the execution of organizational adjustments in reaction to the changing business landscape. According to Weaver [45],

the idea of value creation in the context of project management is made up of two interconnected elements. transforming a concept into something concrete through planned and continuous efforts. The second crucial element entails using creative methods to apply management procedures skillfully in order to effectively control the organization's project control infrastructure.

The strategic objectives of a project are centred on creating and adding value. Nonetheless, an organization's ability to provide value for its clients decides how successful it will be. An organization's values are created and upheld in large part by the PVS role. Recently, there has been an increasing move in many project-based sectors toward customer and stakeholder involvement and co-creation of value.

Its related hypotheses state:

**H7$_o$**: *The project value sustainability role of the project management office is not associated with the implementation of the strategic plan of a project-based organization*

**H7$_a$**: *The project value sustainability role of the project management office is associated with the implementation of the strategic plan of a project-based organization*

## 3. Methodology

The methodology, data collection and analysis adopted by this research are dedicated to figuring out the PMO's roles and responsibilities as functioned in project-based organisations.

This study is explanatory, therefore, the quantitative method was used to gather objective data. The questionnaire contents reflected the aim, theoretical model, research questions, and hypotheses and established a ribbon between them. The online questionnaire was found to be a convenient and cost-effective way to reach more participants, as they were distributed across various geographical locations.

The work of Hobbs, et al. [22] was used as a guide to develop the questionnaire that consists of seven predictor variables and one predicted variable. The questionnaire encompasses a measurement of the roles in the implementation of the strategy in an organization. A purposive sampling was used, which means a target sample population that is involved with the PMO activities in their organizations were selected. The method followed in this research has been discussed in Fig 2.

The questionnaire consists of five parts which include i) the information on demographic information, ii) the services provided in the hosted organization, iii) the execution of the strategic plan of the organization in the presence of the project management office iv) measuring the effectiveness of PMO's role in the hosted organization, and v) qualities to be used as a criteria for evaluation of PMO roles.

The proposed questionnaire is based on a Likert Five-point scale to include the following options ranging from very effective (5) to not effective (1). The scale questions were deliberately constructed to circumvent common survey questionnaire flaws including low response rates and poorly worded questions. However, the number of questions was increased to overcome the flaw of survey-based research of studying a phenomenon in great detail. After getting ethics approval from the UAE university's ethical committee, 450 individuals working in 19 public and semi-public organisations received the online questionnaire link via email (after obtaining written informed consent). The survey ran from April 1, 2020, to July 31, 2020. The companies were chosen based on their significant involvement with PMO activities in the United Arab Emirates. A total of 268 valid questionnaires were obtained, representing a 60% response rate.

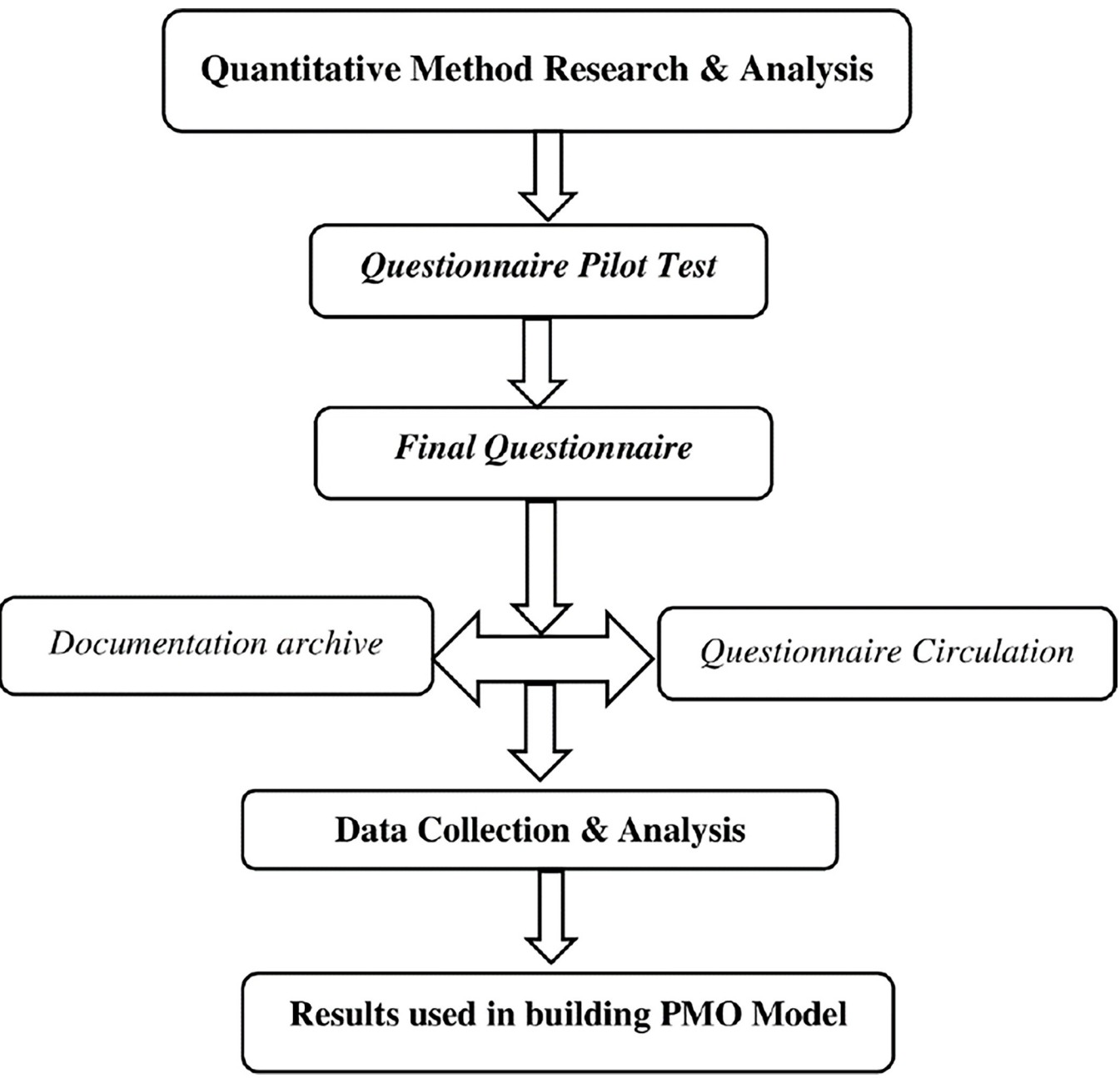

**Fig 2. Quantitative methodology approach (self-created).**

### 3.1 Demographic representation

Project coordinators, managers in the quality department, strategic managers involved with strategic planning, portfolio, program, and project managers, as well as project support specialists (such as IT professionals, economists, engineers, accountants, business analysts, consultants, human resource professionals and contractors, etc.), made up the study sample. The example illustrates a hierarchical management level, from top managers to middle-level project managers and on-site project supervisors. In addition, the participation included both foreign

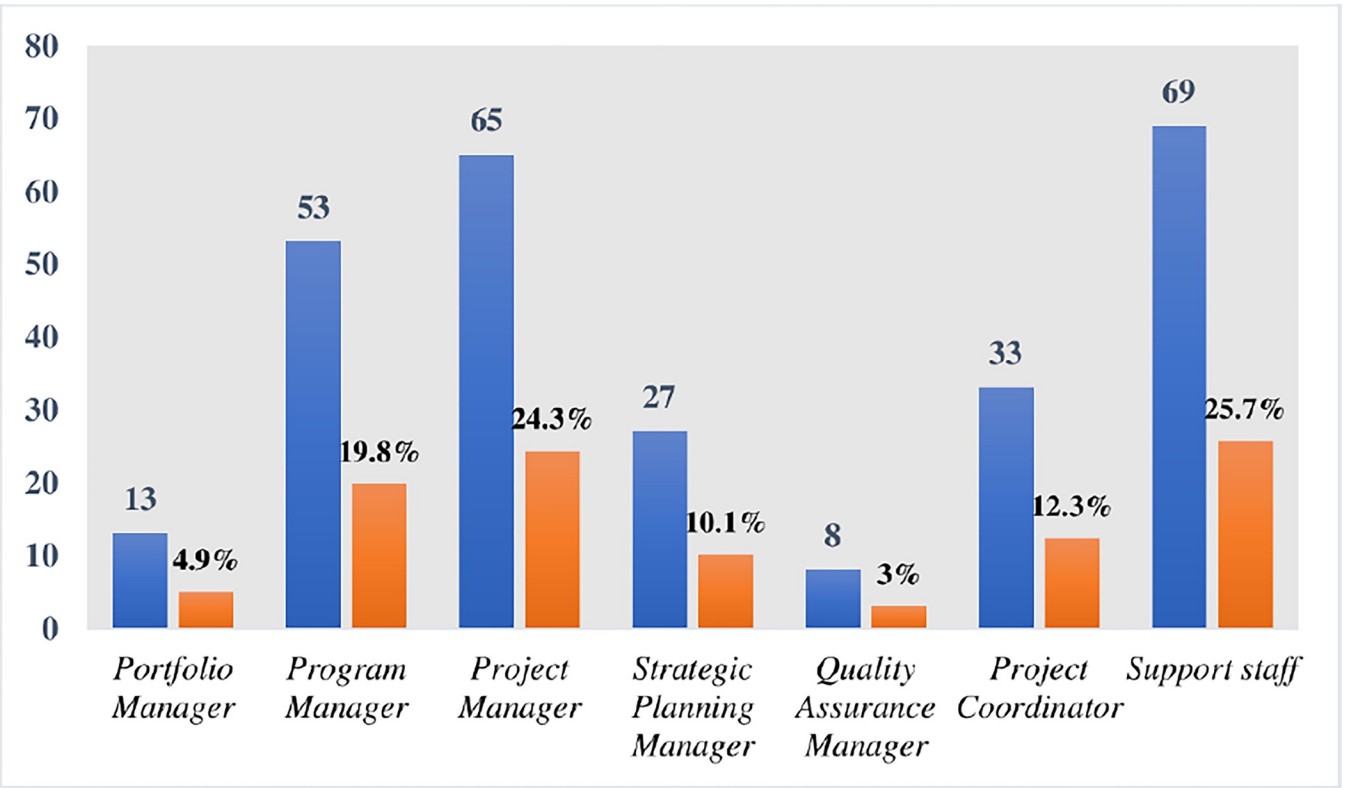

**Fig 3. PM job titles of respondents.**

project workers and Emirati nationals. Since improving an organization's strategic plan is the study's main objective, the organization was utilized as the study's unit of analysis.

The criteria for the selection of the respondents were the academic qualifications, job capacity and years of accumulated experience. The long-term involvement with projects, along with the presence of experienced and literate project managers in these organizations, meant an enhanced ability to answer the main research question. This provided a strong reason to select these organizations. The questionnaire highlighted the importance of the PMO's connection to the host organizations, offering insight into the extent of PMO members' involvement in the operational functions of the affiliated sectors. The demographic data are illustrated in Figs 3–6.

## 4. Results and discussion

What is the role and responsibilities of PMO? The PMO has asked for a crucial step in translating this investigation in order to accomplish its goals and produce and provide significant values. Regression analysis in SPSS was used to statistically analyze the data obtained from the QuartileTM questionnaire.

The seven PMO responsibilities were divided into two distinct tiers, or tactical and strategic roles, with the assistance of the respondents. By classifying the roles according to their functional duties, nature, and patterns of interaction with other predictors in the conceptual model, the efficacy of each role was to be evaluated. The two PMO role categories are strategic role and tactical role. The strategic role consists of Strategic Management (SM); Multi-Project Management (MPM); Organisational Structure and Communication (OSC); Project Value

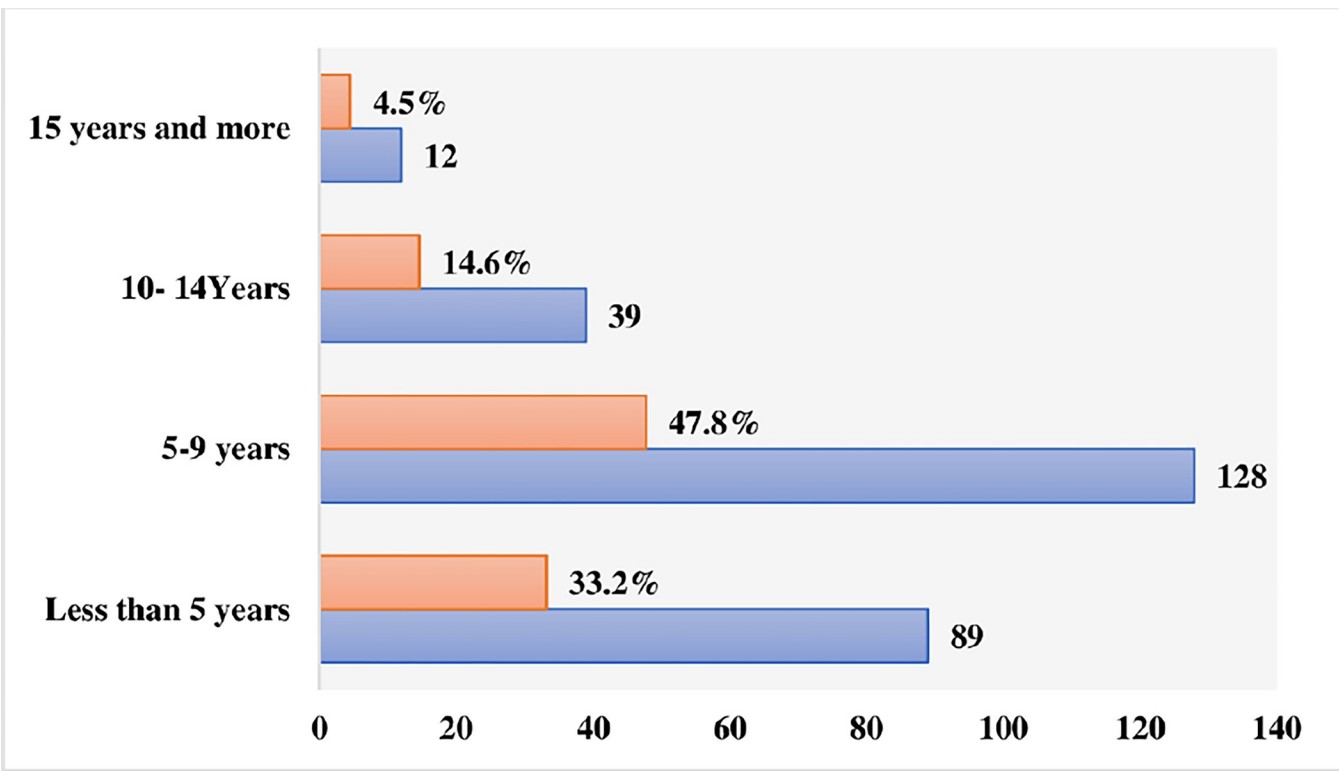

**Fig 4. Years of respondents' experience in PMO-related activities.**

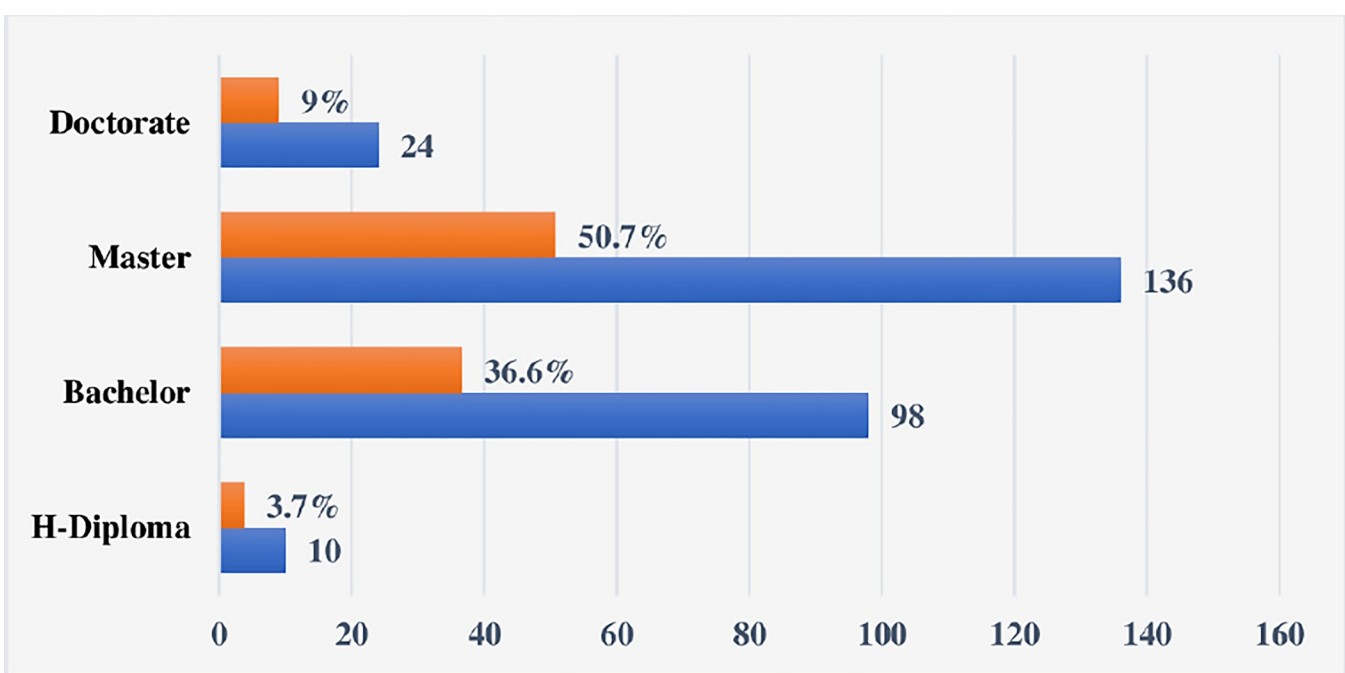

**Fig 5. Academic qualifications of the respondents.**

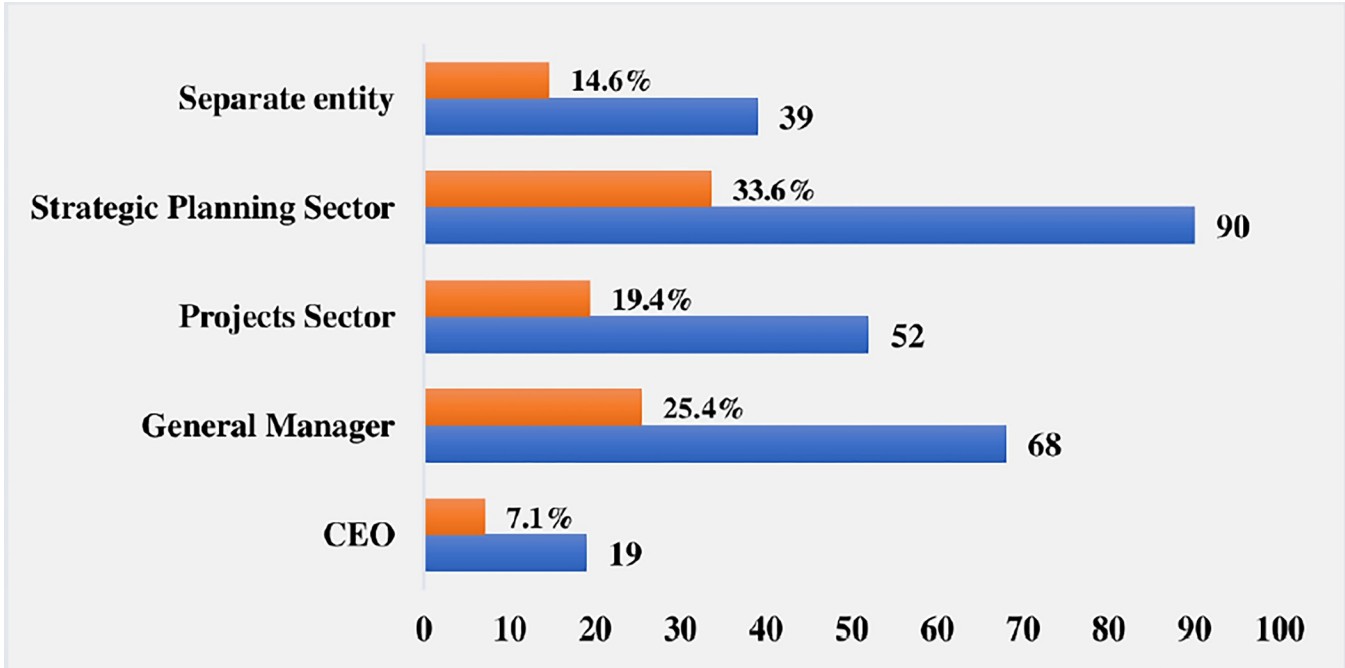

**Fig 6. The PMO members adhered to different organisation's sectors.**

Sustainability (PVS) and the tactical role consists of PM Competency and Methodology (PMMC); PM Monitoring and Controlling (MCP); Organisational Learning Performance (OLP)

The first research question suggested that there are unexplored ways to probe the actual motivation of an organisation in establishing a PMO unit, as a part of its project-based activities. It is likewise, advised that the value of a PMO can be ascertained by analysing its impact within an organisation in terms of the implementation of its strategic plan. The questionnaire inquiries pertaining to the first research question were centred on aspects such as the scope, objectives, roles, line functions, and time considerations essential for the successful execution of the strategic plan. These factors served as motivational drivers, igniting significant interest within organizations to adopt a PMO unit. The Cronbach alpha tests for the potential influence of the PMO roles (individual or confederated) for reaching the organisations' strategic goals through successful execution or even minimising the failure risks of their project business. Cronbach alpha tests for each PMO role were found to be at 0.942, 0.944, 0.946, 0.946, 0.947, and 0.948.

The organization's strategic plan implementation was broken down into factors like meeting the plan's scope, adhering to the organization's suggested goals, staying within the budget, finishing on schedule, earning the trust of stakeholders, and satisfying community needs. The Cronbach alpha was used to test the internal consistency of the six criteria. As previously noted, the Cronbach alpha test resulted in 0.954 for the internal consistency of the successful strategic planning execution criterion, indicating sufficient consistency for this investigation.

The respondents identified specific areas where the PMO unit effectively contributes to the implementation organization's strategic plan. Through the Cronbach alpha test, the results demonstrated a satisfactory response to the first research question regarding the rationale behind establishing a PMO unit. In essence, the PMO fulfils crucial financial functions such as cost and schedule management, supports strategic objectives by ensuring alignment with goals and stakeholders' trust, and engages with the community by addressing their needs and fostering involvement. Moreover, respondents highlighted the PMO's significant potential in

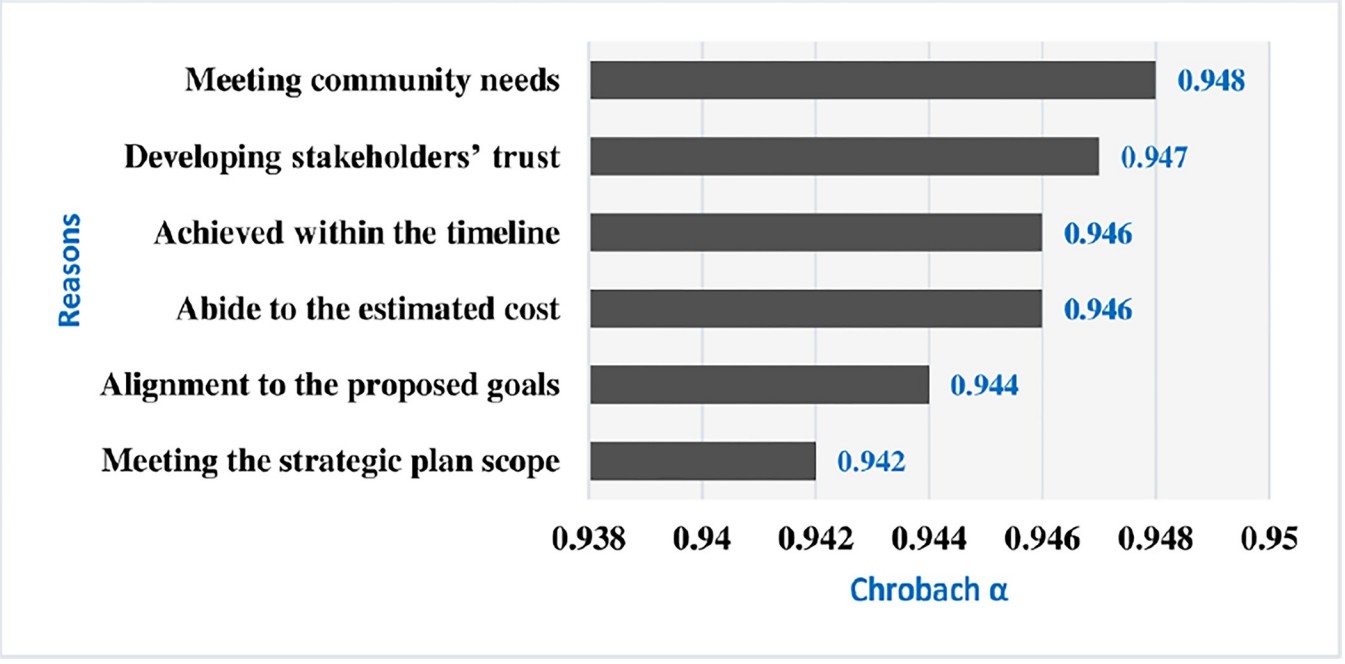

**Fig 7. Cronbach α test for each reason that justifies the PMO establishment.**

meeting the strategic scope and objectives within defined timeframes, underscoring its value-added contributions that validate the establishment of a PMO unit. Top of Form This value-adding is demonstrated in the successful implementation of the projects in the strategic plan context of the hosted organisation.

According to Hobbs, et al. [22], the PMO's current condition in many project-based organizations and organizational project management—which involves carrying out several projects as part of a company's business strategy—are essentially the result of their unique organizational business history. Furthermore, the body of recent PMO research provides strong evidence for the relationship that exists between the strategy of the organization and project operations, highlighting the critical function that the project management methodology plays in facilitating the implementation of the organizational strategy [6] The α test for each of the above reasons for establishing a PMO unit is displayed in Fig 7.

For validity, the KMO test has now been included, in which the result should be between 0.6 and 1 (the better the results, the closer they are to 1). The values in this research ranged from 0.858 to 0.929

Multiple regression analysis was used to identify if the proposed PMO roles had a significant impact on the implementation of the strategic plan of the organization. The goal of the MRA was to determine the possible predictive power of each PMO job on an organization's implementation of its strategic planning. The seven suggested PMO roles' MRA values explained 72.9% of the variance of the SPE construct. The produced MRA data showed how much each PMO role influences the other roles and operates independently. The MRA coefficients of the PMO role indicate the influence and the key parts of each PMO role, as follows:

■ *Strategic Management (SM)*

MRA values of $t = 5.88$, $\beta = 0.374$, and $p < 0.001$ were discovered for the SM's PMO involvement. According to this, SM's PMO involvement was determined to be important, whereas

the alternative hypothesis (H1a) had substantial support. In order to improve project execution and strategic plan implementation, the PMO role of SM was crucial in ensuring the following: i) offering top management advisory services; ii) taking part in strategic planning processes; iii) guaranteeing effective benefits management; and iv) guaranteeing effective environmental scanning.

■ *Project Management Methodology and Competencies* (PMMC)

MRA values of $t = 3.294$, $\beta = 0.234$, and $p < 0.001$ were reported for PMMC's PMO function. According to these findings, PMMC's PMO involvement was determined to be important, and the alternative hypothesis (H2a) received enough support. Enhancing project team competency through professional training, and offering a set of appropriate tools like processes, procedures, templates, etc. were the main components of the PMMC PMO role in enhancing project execution and strategic plan implementation.

■ *Monitoring and Controlling Performance* (MCP)

The MRA values for the MCP role of the project management office were found to be $t = 2.087$, $\beta = 0.158$, and $p = 0.038$. Despite the fact that these data demonstrated that the PMO function of the MCP enhanced the organization's strategic plan implementation, the alternative hypothesis (H3a) is largely supported. The main components of the PMO role of the MCP in improving project execution and strategic plan implementation were as follows: i) reporting project status to top management; ii) monitoring and controlling project performance; iii) implementing and operating project information systems (e.g., Primavera, PMIS, Dashboard, etc.); iv) creating and supporting a project scoreboard; and v) supporting project governance functions.

■ *Organisational Learning Promotion* (OLP)

MRA values of $t = -0.190$, $\beta = 0.012$, and $p = 0.849$ were found for the PMO function of the OPL. These findings indicated that the PMO function of OLP was not significant and that the null hypothesis (H40) should be taken into consideration separately rather than being ruled out. However, the respondents saw the following as critical functions this PMO carried out to improve project execution and strategic plan implementation: Among the activities that must be finished are post-project reviews, project audits, creating and maintaining a database of lessons learned and archived documents, implementing and maintaining a database of project risks, and evaluating the efficacy of the PMO.

A major reason for PMO's organisational learning not supporting the strategy implementation may be the timing of the organisational learning. For example, organizational learning usually takes place after the completion of the project, therefore, its immediate effect on the implementation of the strategic implementation in the same project cannot be substantiated at that time.

*Multi-Project Management* (MPM)

MRA values for the PMO involvement of the MPM were determined to be $t = 0.749$, $\beta = .050$, and $p = 0.455$. The PMO role of the MPM was not shown to be significant, but the null hypothesis (H50) could not be ruled out because this function significantly interacts with other roles. Notwithstanding, the participants highlighted the pivotal functions of the Project Management Office (PMO) of MPM in enhancing project execution and strategic plan implementation. These functions included: i) facilitating coordination amongst ongoing projects; ii) recognizing, evaluating, and assigning priority to new projects; iii) overseeing multiple portfolios and programs; and iv) distributing organizational resources amongst ongoing projects. One of the reasons for the non-significant relationship is the respondents considering a

specific project, while, ignoring the PMO's importance for generic strategy implementation across multiple projects in the firm.

■ *Organisation Structure and Communication* (OSC)

The OSC's PMO role was found to have MRA values of t = 1.978, β = 0.163, and p = 0.049, indicating that it had a significant function with enough evidence for the alternative hypothesis (H6a). The participants identified several pivotal functions of the OSC PMO role in enhancing project execution and strategic plan implementation. These functions included: i) creating a PMO structure that is in synchronized with the needs and objectives of the organization; ii) enhancing communication with project stakeholders; iii) promptly updating project information correspondences; and iv) supporting project continuity by transferring technology and innovative methods.

■ *Project Value Sustainability* (PVS)

MRA values for the PVS's PMO function were determined to be t = -0.651, β = -.047, and p = 0.515. These findings suggested that although the null hypothesis (H70) could not be ruled out, PVS's PMO function was not substantial. However, the respondents emphasized the following crucial roles that PVS's PMO performed in enhancing project execution and strategic plan implementation: Project management for maximum value delivery, ensuring that project outputs align with community needs and social ideals, and providing long-term value to the organization. The final results of the multiple regression analysis are now detailed as follows

$$\mathbf{F(7, 260) = 103.762, p < .0005, R2 = .736}$$

The final multiple regression analysis equation is:

$$\mathbf{Y = 0.357SM + 0.211PMCM + 0.150MCP - 0.012OLP + 0.049MPM}$$
$$\mathbf{+ 157OSC - 0.043PVD + 0.546}$$

It was discovered that four PMO roles—SM, PMMC, PMC, and OSC—had a substantial impact on how well the organizations' strategic plans were implemented. Three PMO responsibilities, however—MPM, OLP, and PVS—were not important. Hobbs and Aubry [21] identified the following five PMO roles had the highest scores: i) development of project management competencies and methodologies; ii) monitoring and controlling project performance; iii) strategic management; iv) multi-project management; and v) organizational learning. On the other hand, our study's top five functions were: 1) "Strategic management" 2) "Developing project management techniques and capabilities" 3) "Tracking and regulating project performance" 4) "Organizational learning" and 5) "Improving communication and organization structure". A comparison between the two findings is shown in Table 2.

**Table 2. Comparison (means) between our research results with the study by Hobbs and Aubry [21].**

| Roles of PMO | Our research | Hobbs and Aubry [21] |
|---|---|---|
| SM | 3.8 | 3.1 |
| PMMC | 3.7 | 3.5 |
| MCP | 3.7 | 3.8 |
| OLP | 3.7 | 3.0 |
| *OSC* | **3.6** | - |
| MPM | 3.6 | 3.2 |
| *PVS* | **3.5** | - |

The difference in the outcomes of these studies in identifying the top five variables may be due to the variation like study. While this study exclusively targeted project-led organizations, primarily in the public sector, the research conducted by Hobbs and Aubry had a global scope, examining the potential roles of PMOs across a diverse range of organizations, predominantly in the private sector. Their study encompassed a broad spectrum of business and industrial activities, exploring various business environments.

The existing PMO literature reviewed for this study did not completely study the role of *Communication Improvement and Organization learning*. Nonetheless, this study demonstrated that the independent variables significantly contributed to the implementation of strategic plans. Given that this investigation marks the initial exploration of this PMO role, it is imperative to validate its efficacy in other project environments across the world through further testing and research. Aubry, et al. [10] support the point that there exists significant variability in the roles, functions, structures, and legitimacy of PMOs across organizations. However, the primary distinctions arise in the structures, roles/functions, and perceived value of PMOs. While multi-project operations offer opportunities to reap strategic management benefits, effectively managing multiple projects within a dynamic business context may necessitate context-specific strategic alignment and measures of value [46].

## 5. Conclusion & future directions

The principal research inquiries focused on investigating the dynamics of the association between the PMO unit and the execution of the strategic plan, which is attained via the accomplishment of effective project and portfolio management. Using multi-regression analysis (MRA), the study investigated how seven different PMO positions and the organization's strategic plan were implemented in relation to one another. The results showed that host organizations' perceptions of the adopted PMO's contributions varied significantly. These differences may be related to the distinct qualities of project-based companies and their particular business plans.

These findings might contribute to the corpus of knowledge in several ways, such as i) by clarifying the pattern of coordination that has emerged between the PMO and other organizations' divisions and providing some understanding of the project business environment. actively involved in implementing the recommended activities within the constraints of the strategic plan of the company. Furthermore, this study aims to bridge the knowledge gap by applying t-test and regression analysis to the responses from the UAE. We argue that improving the way each PMO function is carried out may help organizations become more capable of managing their projects—whether they are individual projects or part of a portfolio—and producing good results.

The theoretical base of the discipline of project management is bleak [47]. However, this has recently started to improve authors such as focusing on the need for theories in the discipline of project management theory [48]. As PMO is a separate entity within an organization, therefore, it can be considered a separate organization [49]. This research would contribute to the project management discipline by further explaining the theory of a project as a temporary organization which claims the contribution of a temporary organisation (a project) to be the overall organizational strategy. As PMO is a separate entity, therefore, this can be used to explain the projects as a temporary organization.

As a practical implication, the findings would fill the void of the unavailability of empirical studies on the role of PMO in implementing the strategic plan and aligning it with the project objectives and outcomes. Therefore, the practitioners can explore and develop their strategic plan more effectively, which shall assist the organization in achieving its goals. The outcome of

the research also has implications for the hosted organizations, such as using PMO, not only as a means of managing projects but also as a tool to improve the strategic plan. As a limitation of the study, and the area of future direction, there is a dire need to explore the implications of PMO practices across different industries/sectors and countries that would provide more rigour and a deeper understanding of the contextual influences on the effectiveness of PMO implementing the strategic plan. This would help improve the strategy implementation in the areas of communication, organizational learning structure etc. Considering the interdisciplinary nature of PMOs, the areas of these future directions can be expanded to other domains such as drawing connections to related fields such as organizational behaviour, information systems, and strategic management. All these disciplines are interlinked with each other and also with project management, which means that this study on PMO can have significant implications in these areas with regard to improvement in performance.

## Supporting information

**S1 Data.**
(XLSX)

## Acknowledgments

Writing this paper has taken a tremendous amount of cooperation from friends near and far, and many anonymous participants. We are indebted to the 268 anonymous respondents to the questionnaire and to the individuals who participated in thoughtful discussions and volunteered with their PMO experience.

## Author Contributions

**Conceptualization:** Maqsood Ahmad Sandhu, Tareq Al Ameri.

**Data curation:** Maqsood Ahmad Sandhu.

**Investigation:** Maqsood Ahmad Sandhu, Tareq Al Ameri.

**Methodology:** Maqsood Ahmad Sandhu, Tareq Al Ameri.

**Supervision:** Maqsood Ahmad Sandhu.

**Validation:** Asjad Shahzad, Afshan Naseem.

**Writing – original draft:** Asjad Shahzad.

**Writing – review & editing:** Maqsood Ahmad Sandhu, Asjad Shahzad, Afshan Naseem.

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
