## [Decision Letter · Decision Letter 0]

22 Apr 2024

PONE-D-24-08102The Role of PMO Practices in the Implementation of Strategic Plans in Project-Based OrganisationsPLOS ONE

Dear Dr. Shahzad,

Thank you for submitting your manuscript to PLOS ONE. After careful consideration, we feel that it has merit but does not fully meet PLOS ONE’s publication criteria as it currently stands. Therefore, we invite you to submit a revised version of the manuscript that addresses the points raised during the review process.

We look forward to receiving your revised manuscript.

Kind regards,

Kashif Ali, PH.D

Academic Editor

PLOS ONE

Journal Requirements:

https://www.emerald.com/insight/content/doi/10.1108/BIJ-03-2018-0058/full/html

In your revision ensure you cite all your sources (including your own works), and quote or rephrase any duplicated text outside the methods section. Further consideration is dependent on these concerns being addressed.

3. We note that your Data Availability Statement is currently as follows: All relevant data are within the manuscript and its Supporting Information files

Additional Editor Comments:

Please address all the comments in detail. 

Reviewers' comments:

Reviewer's Responses to Questions

**Comments to the Author**

1. Is the manuscript technically sound, and do the data support the conclusions?

Reviewer #1: No

Reviewer #2: Yes

Reviewer #3: Partly

Reviewer #4: Yes

2. Has the statistical analysis been performed appropriately and rigorously? 

Reviewer #1: No

Reviewer #2: Yes

Reviewer #3: N/A

Reviewer #4: Yes

3. Have the authors made all data underlying the findings in their manuscript fully available?

Reviewer #1: Yes

Reviewer #2: Yes

Reviewer #3: Yes

Reviewer #4: Yes

4. Is the manuscript presented in an intelligible fashion and written in standard English?

Reviewer #1: No

Reviewer #2: Yes

Reviewer #3: Yes

Reviewer #4: Yes

5. Review Comments to the Author

Reviewer #1: The manuscript titled "The Role of PMO Practices in the Implementation of Strategic Plans in Project Based Organisations" has been critically reviewed. I have some minor suggestions that can enhance the manuscript. It presents a thorough investigation into the impact of Project Management Office (PMO) roles on strategic plan implementation within project-based organizations. I have so many major mistake in this manuscript. It not look like a research paper, because the authors didn’t follow the proper scholarly write-up format. To further enhance the quality and impact of this research work, I have the following comments and suggestions:

Title: The title of paper is very confusing. No need to use Abbreviation in the title.

Abstract: Abstract doesn’t contain relevant information but it contain a lot of extra information which should not be a part of paper abstract.

Clarify the Research Gap: The introduction clearly identifies a gap in the existing literature regarding the strategic role of PMOs. To strengthen this further, it would be beneficial to explicitly state how your study's findings fill this gap and what unique contributions your research makes to the field.

Expand Literature Review: While the literature review is comprehensive, incorporating recent studies could enrich the context. Highlighting studies conducted after 2020 might provide a more current understanding of PMO roles and their evolution in response to emerging project management challenges.

Methodology Detailing: The methodology section is well-structured, but adding more details about the survey design and statistical analysis techniques (beyond regression analysis) could enhance reproducibility and the rigor of your study. Discussing the rationale for choosing the specific quantitative methods and any limitations they present would also be valuable.

Sample: Your sample is robust and geographically diverse, which strengthens the study. However, exploring the implications of PMO practices across different industries or sectors more explicitly could provide deeper insights into how context influences PMO effectiveness in strategic plan implementation.

In-depth Analysis of Non-significant Roles: The results section provides a clear distinction between significant and non-significant PMO roles. An in-depth discussion on why certain roles (e.g., Organisational Learning Promotion) were found to be non-significant and the potential implications for practice could offer additional value to readers.

Practical Implications: The discussion on practical implications is insightful but could be expanded. Providing specific recommendations for project managers and PMO leaders on how to apply your findings in their organizational contexts would make the research more actionable.

Future Research Directions: While the conclusion mentions the potential for future research, outlining specific questions or areas of investigation that emerged from your findings could guide subsequent studies and encourage further exploration into PMO roles and strategic planning.

Theoretical Contributions: Elaborate more on how your findings contribute to or challenge existing theories within project management and strategic planning literature. This would enhance the theoretical significance of your study.

Interdisciplinary Insights: Considering the interdisciplinary nature of PMOs, drawing connections to related fields such as organizational behavior, information systems, and strategic management could enrich the discussion and appeal to a broader audience.

Enhance Visuals and Tables: The figures and tables are informative but ensuring they are as clear and concise as possible will help readers digest the information more effectively. Consider revising to focus on key findings and using supplementary materials for additional details.

Implementing these suggestions could enhance the clarity, depth, and impact of your research, making a valuable contribution to the understanding of PMO roles in strategic plan implementation within project-based organizations.

Reviewer #2: Introduction

• Some points are repeated across different paragraphs, such as the discussion on the challenges faced by project-based organizations and the role of PMO. This repetition can make the text redundant and less engaging.

• Certain statements are vague and could benefit from more specific examples or evidence. For instance, phrases like "unfavourable challenges" and "effective approaches and tools" lack specificity and clarity.

Background

• Some sentences are lengthy and could be simplified for better clarity and readability. For example, the sentence starting with "Many research studies stated..." could be broken down into smaller, clearer sentences.

• Could you please correct the usage of 'PMO' as an abbreviation in two instances: firstly, in the introduction and secondly, in the section discussing the importance of PMO for an organization? Additionally, on line 152, ensure that 'project management office' is spelled out in full, with its abbreviation included. Correct and put abbreviation.

• On page 10, line 216, tackled and gained on" might be more effectively expressed as "addressed and acquired."

• Utilize Strategic human resource management practitioners’ emotional intelligence and affective organizational commitment in higher education institutions in Georgia during post-COVID-19(2023) to enhance your discussion on strategic management. It offers valuable insights on Conceptualization of SM strengthening your argument regarding strategic management.

methodology

• It would be beneficial to provide some insight into the process of composing the questionnaire contents to demonstrate the thoroughness of the methodology.

Other suggestions:

• This study lacks the exploration of both theoretical and practical implications. Please add.

• Please incorporate considerations of limitations and suggest avenues for further research.

Reviewer #3: Dear authors! I have read the manuscript of your article and, while assessing the work done as a whole positively, I have a number of comments:

1. The manuscript is not structured in accordance with the requirements of the PLOS ONE journal. The introduction does not provide a detailed review of the literature and research history on the role of the project management office (PMO). Other parts of the manuscript provide references that were used for the study and the list of references is extensive. A full literature review is required in the introduction.

2. A similar situation has arisen with the description of the methodology and research methods. These should be described in detail in a separate section.

3. When indicating the affiliation of the authors, it is also required to indicate the department, city, and country.

4. The authors conducted a survey among 19 design organizations. However, the justification for the selection of these organizations is very succinct: “The organizations were selected on the basis of their long-term involvement in the activities of the PMO in the UAE.” It is necessary to explain the choice of organizations in more detail, because this influences the formation of further samples.

5. Multiple regression analysis (MRA) was used, I would like to see the resulting regression equations with the described variables, and it is also worth writing the tests that the authors conducted to confirm the quality of the model.

Reviewer #4: Three keywords are really few. You could add two more keywords. Also try to arrange the references in alphabetical order. This will make your work professional. In other words the work is ok but needs small revision

6. PLOS authors have the option to publish the peer review history of their article (what does this mean?). If published, this will include your full peer review and any attached files.

Reviewer #1: No

Reviewer #2: **Yes: **Roya Anvari

Reviewer #3: **Yes: **Julia Kolesnikova

Reviewer #4: **Yes: **Joseph Yaw Dawson

---

## [Author Response · Author response to Decision Letter 0]

10 Jun 2024

To

The Editor in Chief, 

PLOS ONE

Dear Dr Kashif,

First of all, the authors are most grateful to the anonymous referees for their constructive and helpful comments that helped to improve the presentation of the paper considerably. We have carefully noted the reviewers’ feedback and made appropriate changes to the each comment in this revised manuscript. We feel confident that the quality of the manuscript has been significantly improved. 

The following table provides details of our responses to the comments received by the two reviewers. 

Thank you.

Dr Asjad Shahzad

On behalf of authors 

Editor comments – 

 Comment (Originally taken from the feedback letter) Response - All changes are underlined in the revised manuscript. The page and line numbers for the changes refer to the MS Word-generated pages and lines on the revised manuscript.

Please ensure that your manuscript meets PLOS ONE's style requirements, including those for file naming. The PLOS ONE style templates can be found at https://journals.plos.org/plosone/s/file?id=wjVg/PLOSOne_formatting_sample_main_body.pdf and https://journals.plos.org/plosone/s/file?id=ba62/PLOSOne_formatting_sample_title_authors_affiliations.pdf

Thank you very much for bringing our attention to this point. We have now reviewed the formatting style mentioned in the link and have adopted it throughout the main document.

(Throuhgout the Manuscript)

We noticed you have some minor occurrence of overlapping text with the following previous publication(s), which needs to be addressed:

https://www.emerald.com/insight/content/doi/10.1108/BIJ-03-2018-0058/full/html

In your revision ensure you cite all your sources (including your own works), and quote or rephrase any duplicated text outside the methods section. Further consideration is dependent on these concerns being addressed. 

We agree with your point, and in order to address this concern, we checked our submission in Turnitin software, which showed high plagiarism (mostly from our previous work). As advised, we have significantly rephrased the text, which has clearly reduced the overlapping text and plagiarism. We have self-cited the areas which were absolutely necessary. Special focus was given to the text outside the methods section. 

(Throuhgout the Manuscript)

We note that your Data Availability Statement is currently as follows: All relevant data are within the manuscript and its Supporting Information files.

Thank you for bringing our attention to these points. 

We confirm that our submission contains all the raw data required to replicate the results of the study.

Also, the requirements for the ethics statement have been fulfilled in the methods section. 

Moreover, all the other points mentioned in the corresponding left column have been understood and addressed.

 Please include your full ethics statement in the ‘Methods’ section of your manuscript file. In your statement, please include the full name of the IRB or ethics committee who approved or waived your study, as well as whether or not you obtained informed written or verbal consent. If consent was waived for your study, please include this information in your statement as well. 

The requirements for the ethics statement have been fulfilled in the methods section. 

Reviewer 1

 Comment (Originally taken from the feedback letter) Response - All changes are underlined in the revised manuscript. The page and line numbers for the changes refer to the MS Word generated pages and lines on the revised manuscript.

Title: The title of paper is very confusing. No need to use Abbreviation in the title.

Abstract: Abstract doesn’t contain relevant information but it contain a lot of extra information which should not be a part of paper abstract

Thank you for pointing this out. As advised, the abbreviation (PMO) from the title has now been removed, and the revised title is ” Examining the Role of Project Management Offices on Strategic Plan Implementation in Project-Based Organizations” 

With regards to the abstract, extra information deemed irrelevant has now been eliminated especially the one related to unnecessary details mentioning the background of the topic.

Clarify the Research Gap: The introduction clearly identifies a gap in the existing literature regarding the strategic role of PMOs. To strengthen this further, it would be beneficial to explicitly state how your study's findings fill this gap and what unique contributions your research makes to the field. 

Thank you for bringing this point to our attention. We have now attempted to clarify this research gap. The following have now been included in the introduction section

“This research would contribute to the literature by identifying the specific domains that may be improved by adopting the PMO in an organization. Furthermore, this research shall fill the research gap of the unavailability of empirical studies on the role of PMO in implementing the strategic plan. This will help the project-based organizations to align the strategic plan of the organization with the project objectives and outcomes. This shall help the practitioners effectively explore and develop their strategic plan, which in return could assist in achieving organizational goals. Moreover, the study would have significant implications for the hosted organizations, such as utilizing PMO, not only as a means of managing and controlling projects but also as a way to enhance the achievement of the strategic plan. These results can be utilized in developing the conceptual PMO model that would be flexible to be applied to similar project management methodology in various business settings, as well as pave the way for further scholarly investigations”.

(Introduction) 

Expand Literature Review: While the literature review is comprehensive, incorporating recent studies could enrich the context. Highlighting studies conducted after 2020 might provide a more current understanding of PMO roles and their evolution in response to emerging project management challenges. 

As advised, many recent studies conducted after 2020 have now been included in the manuscript. Some of them are as follows. 

1. de Medeiros Junior JV. The contribution of project management offices (PMO) to the strategy implementation in Project-Based Businesses: systematic literature review and proposal of a model. Revista de Gestão e Secretariado. 2021;12(2):301-26

2. Azevedo Junior J, BARROSO ACdO, MONTEIRO CA. An expedited model to appraise project management office value. International Journal of Development Research. 2022

3. Almansoori MTS, Rahman IA, Memon AH, Nasaruddin NAN. Structural Relationship of Factors Affecting PMO Implementation in the Construction Industry. Civil Engineering Journal. 2021;7(12):2109-18

4. Ichsan M, Sadeli J, Jerahmeel G, Yesica Y. The role of project management office (PMO) manager: A qualitative case study in Indonesia. Cogent Business & Management. 2023;10(2):2210359

5. Umasekar V. Evaluating the Role of Project Management Offices (PMOs) in Large-Scale Construction Projects: Insights from Construction Industry Professionals. International Journal of Multidisciplinary: Applied Business and Education Research. 2024;5(1):302-10

(Literature Review)

Methodology Detailing: The methodology section is well-structured, but adding more details about the survey design and statistical analysis techniques (beyond regression analysis) could enhance reproducibility and the rigor of your study. Discussing the rationale for choosing the specific quantitative methods and any limitations they present would also be valuable.

Thank you very much for bringing our attention to the shortcomings in methodology. We have now added more details in the methodology section on the design of the survey and statistical analysis (other than the regression analysis). This has helped in increasing the reproducibility and depth of the study. For example, the following have now been added:

“The initial questionnaire (prototype) has been developed with reference to the work of Hobbs, Aubry (22) . questionnaire consists of 7 independent variables and one dependent variable. The questionnaire encompasses a measurement of the PMO roles involved in the performing the strategic plan of an organization. For the purposes of this study, a target sample population dealing directly or indirectly with the PMO activities within their own organization. The questionnaire consists of five parts i) demographic information, ii) type of the PMO services in the hosted organization, iii) execution of the organization’s strategic plan in the presence of PMO entity, iv) measuring the effectiveness of suggested PMO roles in the hosted organization, and v) selection of attributes that could be used as criteria for the evaluation of PMO roles, in general”

Moreover, the rationale of choosing specific quantitative methods and the limitations it presents have now been articulated in the following words:

“The idea of formulating precise written questions for those whose opinions or experience you are interested in seems such an obvious strategy for finding the answers to the issues that are of great interest Creswell (2006)

The scale questions were deliberately constructed to circumvent common survey questionnaire flaws including low response rates and poorly worded questions. However, the number of questions was increased to overcome the flaw of survey-based research of studying a phenomenon in great detail”.

(Research methodology) 

Sample: Your sample is robust and geographically diverse, which strengthens the study. However, exploring the implications of PMO practices across different industries or sectors more explicitly could provide deeper insights into how context influences PMO effectiveness in strategic plan implementation. 

Thank you for appreciating the robustness and diversity of the sample. We have now added this as a limitation of the study in the words below so that future research can be conducted to address this shortcoming.

“As a limitation of the study, and the area of future direction, there is a dire need to explore the implications of PMO practices across different industries/sectors and countries that would provide more rigour and deeper understanding of the contextual influences on the effectiveness of PMO in the implementation of the strategic plan”. (Conclusion) 

In-depth Analysis of Non-significant Roles: The results section provides a clear distinction between significant and non-significant PMO roles. An in-depth discussion on why certain roles (e.g., Organisational Learning Promotion) were found to be non-significant and the potential implications for practice could offer additional value to readers. Thank for the comment. Now the non-significant roles, such as organisational learning promotion, and Muli-project management have been explained in detail. Some of the explanations are in the following words:

“A major reason for PMO’s organisational learning not supporting the strategy implementation may be the timing of the organisational learning. For example, the organizational learning usually takes place after the completion of the project, therefore, its immediate effect on the implementation of the strategic implementation in the same project cannot be substantiated at that time”.

(Result and Discussion) 

Practical Implications: The discussion on practical implications is insightful but could be expanded. Providing specific recommendations for project managers and PMO leaders on how to apply your findings in their organizational contexts would make the research more actionable. 

Thank you. The practical implications have now been discussed in greater detail and rigour. Some of it is detailed as follows:

“As a practical implication, the findings would fill the void of unavailability of empirical studies on the role of PMO in implementing the strategic plan and aid the project-based organizations to align the strategic plan of the organization with the project objectives and outcomes. Therefore, the practitioners can explore and develop their strategic plan more effectively, which shall assist in achieving the goals of the organization. The outcome of the research also has implications for the hosted organizations, such as using PMO, not only as a means of managing projects but also a tool to improve the strategic plan”.

(Conclusion) 

Future Research Directions: While the conclusion mentions the potential for future research, outlining specific questions or areas of investigation that emerged from your findings could guide subsequent studies and encourage further exploration into PMO roles and strategic planning 

Thank you for mentioning this. The future directions have now been made much more specific. Some of the excerpts from the newly added text in the manuscript are as follows:

“In the area of future direction, there is a dire need to explore the implications of PMO practices across different industries/sectors and countries that would provide more rigour and deeper understanding of the contextual influences on the effectiveness of PMO in the implementation of the strategic plan. This would help improve the strategy implementation in the areas of communication, organizational learning and structure etc”.

(Conclusion)

Theoretical Contributions: Elaborate more on how your findings contribute to or challenge existing theories within project management and strategic planning literature. This would enhance the theoretical significance of your study. 

Thank you for bringing our attention towards to need for theoretical contribution in our paper. We have reviewed numerous theories in the field of project and strategic management and attempted to develop a link with our findings. For example.

“The theoretical base of the discipline of project management is bleak [26]. However, this has recently started to improve authors such as focusing on the need for theories in the discipline of project management theory[27]. As PMO is a separate entity within an organization, therefore, it can be considered a separate organization[28]. This research would contribute to the project management discipline by further explaining the theory of a project as a temporary organization which claims the contribution of a temporary organisation (a project)to be the overall organizational strategy. As PMO is a separate entity, therefore, this can be used to explain the projects as a temporary organization”.

(Conclusion)

Interdisciplinary Insights: Considering the interdisciplinary nature o

---

## [Decision Letter · Decision Letter 1]

23 Jun 2024

The Role of Project Management Office in the Implementation of Strategic Plans in Project-Based Organisations

PONE-D-24-08102R1

Dear Dr. Shahzad,

We’re pleased to inform you that your manuscript has been judged scientifically suitable for publication and will be formally accepted for publication once it meets all outstanding technical requirements.

Kind regards,

Kashif Ali, PH.D

Academic Editor

PLOS ONE

Additional Editor Comments (optional):

Reviewers' comments:

Reviewer's Responses to Questions

**Comments to the Author**

1. If the authors have adequately addressed your comments raised in a previous round of review and you feel that this manuscript is now acceptable for publication, you may indicate that here to bypass the “Comments to the Author” section, enter your conflict of interest statement in the “Confidential to Editor” section, and submit your "Accept" recommendation.

Reviewer #2: All comments have been addressed

Reviewer #4: All comments have been addressed

2. Is the manuscript technically sound, and do the data support the conclusions?

Reviewer #2: Yes

Reviewer #4: Yes

3. Has the statistical analysis been performed appropriately and rigorously? 

Reviewer #2: Yes

Reviewer #4: Yes

4. Have the authors made all data underlying the findings in their manuscript fully available?

Reviewer #2: Yes

Reviewer #4: Yes

5. Is the manuscript presented in an intelligible fashion and written in standard English?

Reviewer #2: Yes

Reviewer #4: Yes

6. Review Comments to the Author

Reviewer #2: All comments answered by the authors

Please use the space provided to explain your answers to the questions above. You may also include additional comments for the author, including concerns about dual publication, research ethics, or publication ethics. (Please upload your review as an attachment if it exceeds 20,000 characters) (Limit 100 to 20000 Characters)

Reviewer #4: The authors have made necessary changes to the article though not all the concerns were done. In any case they have improved the work

7. PLOS authors have the option to publish the peer review history of their article (what does this mean?). If published, this will include your full peer review and any attached files.

Reviewer #2: No

Reviewer #4: **Yes: **Joseph Yaw Dawson

---

## [Editor Report · Acceptance letter]

9 Jul 2024

PONE-D-24-08102R1 

PLOS ONE

Dear Dr. Shahzad, 

I'm pleased to inform you that your manuscript has been deemed suitable for publication in PLOS ONE. Congratulations! Your manuscript is now being handed over to our production team.

Kind regards, 

on behalf of

Dr. Kashif Ali 

Academic Editor

PLOS ONE